# Autophagy: New Insights into Mechanisms of Action and Resistance of Treatment in Acute Promyelocytic leukemia

**DOI:** 10.3390/ijms20143559

**Published:** 2019-07-20

**Authors:** Mohammad Amin Moosavi, Mojgan Djavaheri-Mergny

**Affiliations:** 1Department of Molecular Medicine, National Institute of Genetic Engineering and Biotechnology, Tehran 14965/161, Iran; 2Equipe labellisée par la Ligue contre le cancer, Université Paris Descartes, Université Sorbonne Paris Cité, Université Paris Diderot, Sorbonne Université, INSERM U1138, Centre de Recherche des Cordeliers, Paris 75006, France; 3Metabolomics and Cell Biology Platforms, Institut Gustave Roussy, Villejuif 94805, France

**Keywords:** Acute promyelocytic leukemia, all-*trans* retinoid acid, arsenic trioxide, autophagy, cell survival, differentiation, resistance to therapy

## Abstract

Autophagy is one of the main cellular catabolic pathways controlling a variety of physiological processes, including those involved in self-renewal, differentiation and death. While acute promyelocytic leukemia (APL) cells manifest low levels of expression of autophagy genes associated with reduced autophagy activity, the introduction of all-*trans* retinoid acid (ATRA)—a differentiating agent currently used in clinical settings—restores autophagy in these cells. ATRA-induced autophagy is involved in granulocytes differentiation through a mechanism that involves among others the degradation of the PML-RARα oncoprotein. Arsenic trioxide (ATO) is another anti-cancer agent that promotes autophagy-dependent clearance of promyelocytic leukemia retinoic acid receptor alpha gene (PML-RARα) in APL cells. Hence, enhancing autophagy may have therapeutic benefits in maturation-resistant APL cells. However, the role of autophagy in response to APL therapy is not so simple, because some autophagy proteins have been shown to play a pro-survival role upon ATRA and ATO treatment, and both agents can activate ETosis, a type of cell death mediated by the release of neutrophil extracellular traps (ETs). This review highlights recent findings on the impact of autophagy on the mechanisms of action of ATRA and ATO in APL cells. We also discuss the potential role of autophagy in the development of resistance to treatment, and of differentiation syndrome in APL.

## 1. Introduction

### 1.1. Acute Promyelocytic Leukemia 

Acute myeloid leukemia (AML) encompasses a class of hematological diseases that represent an arrest of myeloid precursors at different stages of differentiation [1,2]. Acute promyelocyte leukemia (APL) accounts for about 10% of all AML cases and is characterized by a clonal expansion associated with a blockade of the terminal differentiation of promyelocytes into granulocytes. APL is mostly caused by a reciprocal chromosomal translocation between the retinoic acid receptor alpha gene (RARα) and promyelocytic leukemia (PML), and this affects a number of cellular processes, including senescence, RNA processing and apoptosis [3,4]. The resulting chimeric PML-RARα retains the DNA and ligand-binding domains of RARα. PML-RARα is assumed to be the critical oncogenic event that drives APL pathologies [3]. Two main mechanisms are proposed for this oncogenic function; they are based on the ability of PML-RARα to repress both the PML/p53-driven senescence program through disruption of PML nuclear bodies and the epigenetic transcription of the retinoic acid target genes [5] (Figure 1). These target genes are involved in self-renewal or differentiation following recruitment of transcription co-repressor complexes (COR). Yet, PML-RARα-driven APL development in murine tumors requires secondary oncogenic events, such as Wilms’ tumor 1 (WT1), KRAS, NRAS mutations, and fms-like tyrosine kinase 3 (FLT3) activation [6]. 

### 1.2. The Modes of Action of APL Therapy 

The great majority of acute promyelocytic leukemia patients (95%) are successfully cured in clinical settings after introduction of two agents that both bind and directly target PML-RARα: all-trans retinoic acid (ATRA); and arsenic trioxide (ATO) [7]. In this section, we briefly summarize the molecular mechanisms of action and resistance to ATRA and ATO in APL cells [7,8,9]. 

### 1.3. All-Trans Retinoic Acid (ATRA)

The activity of ATRA is primarily mediated by its binding to the nuclear receptors retinoid X receptor (RXR) and RARα [10]. These receptors bind as homodimers or heterodimers to a specific region of DNA called retinoic acid response elements (RAREs) (Figure 1). In addition to its transcriptional activity, RARα has other functions and can directly activate several signaling pathways [10]. 

In the absence of retinoid acid, RXR and RARα recruit co-repressors (silencing mediator of retinoic acid and thyroid hormone receptor (SMRT)/nuclear receptor corepressor (NCoR1) associated with histone deacetylases, leading to the transcription repression of target genes. Following binding of retinoid acid to their cognate receptors, the co-repressor-histone deacetylase complex is dissociated and the transcription co-activator complexes are recruited to RAREs, allowing the transcriptional activation of different sets of genes. In APL, PML-RARα disrupts ATRA signaling in a dominant negative manner by enhancing binding of transcriptional repressor complexes. In this context, only the addition of pharmacological concentrations of ATRA (10^−6^ M) (i.e., higher than the physiological concentration) can trigger transcriptional de-repression, and this occurs through dissociation of the transcriptional co-repressors and activation of co-activators [11,12,13]. 

A body of evidence has revealed that ATRA also acts on the stability of the PML-RARα protein by inducing its degradation through the proteasome and autophagy pathways-dependent mechanisms [14] (Figure 1). Thus, ATRA has dual therapeutic effects in APL: transcription activation of differentiation genes; and degradation of the PML-RARα oncoprotein. However, some patients are refractory to initial ATRA treatment or relapse due to acquired resistance to ATRA after prolonged treatment and/or incomplete clearance of leukemia-initiating cells residing in bone marrow. 

Both clinical and cellular analyses revealed that resistance to retinoid acid therapy is caused primarily by genetic mutations (deletion, missense and nonsense) in the ligand-binding domain of PML-RARα [15]. These genetic changes may reduce the affinity of ATRA and prevent the release of transcription repressor complexes even under pharmacological doses of ATRA. 

Recently, the existence of mutations associated with ATRA resistance has been shown through whole exome sequencing analysis of diagnosis and relapsed APL patients [16,17]. These studies revealed that relapsing APL patients harbor at diagnosis high incidence of mutations affecting epigenetic or transcriptional regulators (such as WT1, PML and RARA), as well as AT rich interactive domain 1B (ARID1B), AT rich interactive domain 1A (ARID1A) and the MAP kinase pathway. These mutations that are mostly preserved in the relapse APL clones are probability involved in the emergence of other oncogenic alterations or mutations that ultimately impede therapy response. 

### 1.4. Arsenic Trioxide

Arsenic trioxide (ATO) is commonly used, alone or in combination with ATRA, for the treatment of APL. ATO has no therapeutic potential on different subtypes of APL (PLZF-RARα–driven APL), due to the fact that the therapeutic efficiency of this drug is dictated by its binding to the PML moiety of PML-RARα [18,19]. In normal cells, PML operates as a tumor suppressor protein through inducing the formation of nuclear PML bodies, which ultimately leads to p53-driven senescence program. The expression of PML-RARα in APL cells disrupts the formation of nuclear PML bodies, and this consequently impairs the senescence onset [3,20]. The administration of ATO in APL cells causes oxidation, sumoylation and subsequently poly-ubiquitination and proteasomal degradation of the PML-RARα protein (Figure 1), leading to the re-formation of nuclear bodies and the induction of senescence [21,22,23]. ATO can also induce apoptotic cell death and/or partial differentiation, depending on the concentration used [23,24]. At low concentrations (0.1–0.5 µmol/L), ATO promotes differentiation of APL cells, while at higher concentrations (0.5–2 µmol/L), it activates the intrinsic apoptotic cell death (mitochondrial) pathway [24]. Therefore, the mechanisms of action of ATO and ATRA are different in APL cells harboring PML/RARα. ATRA stimulates terminal differentiation of promyelocytes, and conversely, amplification (proliferation and self-renewal) of the stem cells, while ATO induces both apoptosis and differentiation in APL cells without any cytotoxic effect on stem cells [25]. Recently, new links between ATO and other types of cell death (autophagy and ETosis) have been also described in APL cells, which will be discussed later in this section [26]. These cumulative anti-leukemic effects of ATO eventually lead to the eradication of leukemia-initiating cells, which explain why this drug is highly effective as a single agent in APL patients (80%) while ATRA monotherapy causes relapsed disease in the majority of patients. In several clinical settings, the combination of ATO with ATRA improved the outcome of refractory and relapsed patients, making this treatment regimen the most effective therapy for APL patients. In contrast to ATRA, little is known of the mechanism(s) of resistance to ATO. So far, several mutations that lead to substitution of amino-acids in the PML domain of PML-RARα and affect ATO binding sites have been identified in APL patients who are resistant to chemotherapy [21,27,28]. Alex et al. used the whole exome sequencing technique to understand ATO-resistance mechanisms in APL and identified alterations in the redox system, the ubiquitin-proteasome degradation pathway and the PI3-AKT signaling pathway [27]. Consistent with this report, the lower levels of ROS, glutathione and glucose uptake have been observed in ATO-resistant NB4 cells, proposing metabolic rewiring hypothesis as ATO-resistance mechanism in APL cells [29]. Furthermore, a recent report provided evidence for the involvement of microenvironment-mediated drug resistance in ATO-treated APL cells, which is driven by the nuclear factor kappa B (NF-kappa B) pathway [30]. Generally, ATO at therapeutic concentration is well tolerated as its long-term toxicity is mild. However, several side effects, including cardiac and liver damages, as well as differentiation syndrome, have been reported in the patients who underwent chronic ATO therapy, as a single agent or in combination with ATRA [22,31]. 

### 1.5. Differentiation Syndrome 

Differentiation syndrome (DS), previously named retinoic acid syndrome, is a common life-threatening complication in patients with APL who are treated with ATRA and/or ATO [32,33]. The incidence of DS for patients treated with ATRA ranges from 2% to 31%, while this range for patients treated with ATO is wider (7% to 60%). The incidence of DS induced by a combination of ATO and ATRA is between 14% and 25% [34,35]. The onset of DS is initiated by an increase in white blood cells (hyperleukocytosis) and patients present with the prominent signs of unexplained fever, respiratory distress, weight increase, lower-extremity edema, dyspnea, pleural or pericardial effusions, hypotension, and/or acute renal failure [36,37,38]. 

The physiopathology and diagnosis of DS are complex and still remain largely unknown [39]. However, it is believed that ATRA enhances the expression levels of cell adhesion molecules and promotes an excessive systemic inflammatory response, which finally leads to endothelial cell damage, capillary permeability, vessel occlusion, and massive tissue infiltration of differentiating APL cells [32,39,40]. The hematopoietic growth factors, such as interleukin (IL)-1β, IL-6, IL-8, and tumor necrosis factor (TNF), play a central role in DS through promoting leukocyte activation and the binding of APL cells to endothelium [39,41]. Another molecular mediator for the development of DS is cathepsin G, which increases capillary permeability and endothelium damage [42]. The optimal therapeutic strategy for treatment of DS has not been yet established [43]. Corticosteroids are commonly used as a preventive strategy for DS [32,44]. For example, the early administration of dexamethasone (5–10 mg twice intravenously daily) is promising for DS prophylaxis [43,45]. However, in severe DS cases (respiratory or acute renal failure), the interruption of ATRA and/or ATO treatment is mandatory [44,46].

## 2. The Autophagy Machinery and its Role in Cancer 

Macroautophagy (hereafter referred to as autophagy) is a highly conserved lysosomal-dependent catabolic process [47,48]. During autophagy, various intracellular materials including protein aggregates and damaged organelles are sequestered within double-membrane vesicles (called autophagosomes) and then delivered to lysosome for degradation and recycling. Autophagy is orchestrated by a set of autophagy-related genes (ATGs) that are under the control of several signaling pathways, including the mechanistic target of rapamycin kinase (MTOR) and AMP-activated protein kinase (AMPK) axes [47,49,50] (Figure 2). The autophagy process occurs through a multistep process involving initiation, elongation and maturation (Figure 2). The initiation step is the formation of the initiation membrane or phagophore that requires the assistance of two complexes, the unc-51 like autophagy activating kinase 1 (ULK1)/focal adhesion kinase family interacting protein of 200 kD (Fip200) complex and the class II phosphoinositide 3-kinase (PI3K)/BECN 1 complex. This results in the generation of phophatidyl inositol 3 phosphate, which subsequently binds to WD-repeat protein interacting with phosphoinositides (WIPI) proteins effectors, allowing the recruitment of several multiprotein complexes: ATG5-ATG12-ATG16L and the ATG8 family (LC3 or gamma-aminobutyric acid receptor-associated protein (GABARAP)) conjugated with phosphatidylethanolamine to the autophagosomal membrane. This step is cardinal for the expansion and closure of the autophagosomal membrane. The last step is the degradation of the contents of the autophagosome by the lysosomal acid hydrolases upon autophagosome–lysosome fusion, which allows the production of energy and recycling of biomaterials (e.g., amino acids, free fatty acids and monosaccharides) [51,52]. In addition to ATG proteins, selective autophagy requires a subset of proteins, named autophagic adapter proteins, which recognize the cargoes (e.g., mitochondria, organelles, lipid droplets) and deliver them to the autophagosome through their interaction with LC3/GABARAP via conserved W/F/YxxL/I/V [53,54]. The adaptor proteins can either directly bind to the cargoes as was seen for the adaptor proteins NIX and family with sequence similarity 134 member B (FAM134B) or interact with ubiquitinated cargoes though their ubiquitin binding domain as exemplified for p62/sequestosome 1 (SQSMT1), next to BRCA1 gene 1 protein (NBR1), Optineurin and nuclear dot protein 52 kDa (NDP52) [55]. A body of evidence indicates that autophagy plays a critical role in long-lived and non-replicative cells such as quiescent cells as well as terminally differentiated cells [56]. Indeed, autophagy is essential for proper hematopoesis and its deregulation is associated with several hematological diseases including leukemia and lymphoma [57,58]. 

The relationship between autophagy and cancer is complex and depends on the type and stage of tumors [48,59]. It is well documented that both deficiency and overactivation of autophagy can contribute to the development and progression of cancers. On the one hand, autophagy functions in tumor suppression by controlling cell proliferation, differentiation as well as sustaining genome stability and cell homeostasis of normal cells; its inhibition thus contributes to the acquisition of a malignant phenotype at early stages of tumorigenesis. On the other hand, autophagy is reactivated at the advanced stages of malignancy to provide metabolic needs and promote tumor growth, invasion and metastasis. Furthermore, a body of evidence revealed that autophagy controls cancer therapeutics in both positive and negative ways [60,61]. Therefore, depending on the cancer type and context, both inhibition and stimulation of autophagy have been proposed as potential approaches for the treatment and prevention of cancers [59,62,63].

## 3. Regulation of Autophagy in APL Cells at Basal Level and in Response to Treatment

The deregulation of constitutive autophagy in APL cells was revealed by clinical data showing that primary blast cells of AML patients exhibit frequently lower expression levels of a variety of ATG genes (i.e., ULK1, BECN1, ATG14, ATG5, ATG7, ATG3, ATG4B and ATG4C) as compared to mature granulocytes from healthy donors [12,64] (Figure 3). These observations are found irrespective of the genetic abnormality types and the karyotypes of AML samples, including APL primary blast cells. In the same vein, analysis of blast cells of two APL patients showed a significant increase in the expression of the ATG5 gene following introduction of ATRA [64]. The PU-1 transcription factor was shown to be responsible for the transcription induction of several ATG genes during ATRA-induced differentiation of APL cells. MicroRNA-106a was also identified as a negative regulator of ULK1 expression levels in APL cells [64]. Beside the autophagy machinery, we found that the mRNA expression levels of the autophagy adapter p62/SQSTM1 are significantly downregulated in AML patient samples relative to terminally differentiated granulocytes from healthy donors [65,66]. We showed that the activation of the NF-kappaB transcription factor is required for p62/SQSMT1 upregulation during granulocytic differentiation of APL cells. Moreover, the p62/SQSMT1 expression levels are also targeted by several microRNA, including miR-17, -20, -93, and -106 that are expressed at higher levels in hematopoietic blast cells than in mature myeloid cells [67]. Interestingly, high expression of p62/SQSTM1 was associated with poor prognosis in human AML and loss of p62/SQSMT1 in murine in vivo models of AML impairs leukemia progression and mitophagy, supporting the idea that p62 /SQSMT1 plays a pro-leukemic effect in AML through an autophagy/mitophagy-dependent mechanism [68]. 

Altogether, the abovementioned data reveal that APL cells manifest low autophagy gene expression levels associated with reduced autophagy activity, which may favor preleukemia development and promote full leukemia transformation through cooperation with some oncogenic alterations (e.g., PML-RARα). It is worth noting that the expression of mRNA levels of ATG3, ATG4D, ATG5 and p62/SQSMT1 are also increased upon in vitro granulocytic differentiation of primary CD34+ progenitors, supporting the idea that upregulation of autophagy-related genes are common features of granulocytes/neutrophil differentiation in both normal and APL blasts [64,65,66]. In fact, a recent study reported that autophagy is required for metabolic adaptation and differentiation in normal neutrophils [69].

In APL cells, PML-RARα was reported to repress PU-1-dependent transcription activation of a broad range of genes including those involved in autophagy, underscoring the inhibitory effect PML-RARα on autophagy [64,70]. A recent study provided evidence for a new function for PML as a repressor of autophagy when associated at the mitochondrial-associated membrane (MAM) [6]. However, whether or not PML-RARα inhibits autophagy through its binding to MAM has not yet been investigated. These observations differ however from data reporting that PML-RARα is involved in constitutive activation of autophagy in APL cells [71]. The reason for seemingly opposite effects of PML-RARα on the regulation of autophagy remains to be clarified.

## 4. Regulation of Autophagy during Differentiation of APL Cells by ATRA 

We and others provided evidence that ATRA promotes the accumulation of autophagosomes accompanied with the activation of the autophagy flux in APL cells (Figure 3). These responses are associated with increased expression levels of several autophagy-regulatory proteins, including BECN1, microtubule-associated protein 1S (MAP1S), DNA damage regulated autophagy modulator 1 (DRAM1), high–mobility group box 1 (HMGB1) and p62/SQSMT1, as well as downregulation of pathways that repress autophagy (i.e., MTOR and B-cell lymphoma 2 (BCL-2)) [12,64,66,72,73].

The activation of autophagy in mature APL cells depends on a mechanism that involves WIPI but not BECN1, suggesting that the complete autophagy machinery is dispensable for differentiation-associated autophagy [70,72]. Pharmacological activation of autophagy by rapamycin sensitizes APL cells to ATRA-induced differentiation, while silencing of WIPI and ATG7 attenuates granulocytic differentiation, suggesting that autophagy elements (at least some ATGs) contribute to the differentiation-promoting effect of ATRA [12,64]. Interestingly, we found that ATRA-induced autophagy occurs in maturation-sensitive APL cells but not in maturation-resistant cells, linking autophagy defect to differentiation resistance [66]. Similarly, ATRA-induced p62/SQSTM1 upregulation is observed in maturation-sensitive APL cells but not in maturation-resistant cells, supporting the relationship between p62/SQSMT1 upregulation and differentiation [66]. Interestingly, when differentiation was reestablished in maturation-resistant APL cells (by addition of 8-CPT-cAMP to ATRA treatment), autophagy activation and p62/SQSTM1 upregulation were both restored in these cells, underscoring the idea that autophagy and differentiation are interconnected processes. Furthermore, matrine, an alkaloid found in plants from the genus Sophora, was shown to improve the differentiation ability of ATRA-resistant APL cells through induction of autophagy and ubiquitin degradation of PML-RARα [74]. In another study, the addition of lithium (an autophagy inducer) to ATRA reactivated differentiation in ATRA maturation-resistant HL60-Diff-R AML cells, emphasizing the concept that autophagy modulation may represent a means to restore differentiation [12]. One proposed mechanism for the role of autophagy in differentiation relies on the selective degradation of the PML-RARα oncoprotein by autophagy, which allows the release of differentiation blockade in APL cells [14]. This occurs through several mechanisms: one study revealed the involvement of the long-noncoding RNA (lncRNA) HOTAIRM1 in the induction of autophagy and the degradation of PML-RARα. [75]. Another study showed that miR125B1 targets DRAM2 (a key regulator of autophagy) leading to inhibition of PML-RARα degradation and differentiation [76]. It has also been shown that in response to ATRA, the autophagy adapter p62/SQSMT1 interacts with PML-RARα through its ubiquitin binding domain, resulting in PML-RARα degradation and differentiation activation [77]. In addition, PML-RARα was also shown to interact with autophagy-linked FYVE-domain containing protein (ALFY), an autophagy scaffold protein that assists p62 /SQSMT1 in the autophagy-dependent degradation of ubiqutinated protein aggregates [78]. HMGB1 is another regulator of autophagy involves in PML-RARα degradation in APL cells although the mechanism underlying this regulation is not fully understood [73]. Knockdown of p62/SQSMT1 increased the abundance of ubiquitinated protein aggregates in APL cells subjected to ATRA treatment, suggesting that PML-RARα is not the sole protein degraded by the p62/autophagy pathway [66]. Besides clearing proteins, autophagy was shown to be involved in selective lipid degradation during normal neutrophil differentiation [69]. Whether or not such selective autophagy occurs during the course of ATRA-induced granulocytic differentiation of APL cells remains an open question. 

Another mechanisms through which autophagy promotes APL differentiation relies on the modulation of the NEDD8 E1 axis. Indeed, inhibition of neddylation by MLN4924 (a small molecule that specifically blocks the activity of NEDD8 E1-activating enzyme) in APL cells resulted in autophagy activation, which consequently enhanced ATRA-induced differentiation of APL cells [79]. 

It is worth nothing that ATRA is also able to promote autophagy and differentiation in myeloid leukemia HL60 cells that do not express PML-RARα, supporting the idea that PML-RARα degradation is not the sole mechanism through which autophagy promotes differentiation in response to ATRA [64]. Another possibility is that ATRA-induced autophagy provides metabolites and energy supply for differentiation engagement, as shown for normal neutrophil maturation [69]. In fact, the identification of additional mechanisms involved in this process will lead to a better understanding of the regulation of myeloid differentiation by autophagy. 

Finally, little is known about the individual role of each autophagy-regulatory protein in the control of cell death and the self-renewal capacity of APL cells upon ATRA treatment. In this regard, we found that, unlike WIPI and ATG7, BECN1 and p62/SQSMT1 are not involved in granulocyte differentiation of APL cells, but rather protect cells against death, suggesting their pro-survival role under this condition [12,66,70,72]. In another study, it was reported that ATG5 knockdown following ATRA treatment causes APL cell death while limiting granulocytic differentiation, suggesting a distinct role of ATG5 in regulation of cell fate relative to other ATG proteins [64]. 

Altogether, these findings support the idea that ATG proteins play distinct roles in APL cell fate during ATRA-induced granulocytes differentiation. This observation should be taken into account when autophagy modulation, alone or in combination with other drugs, is considered for the treatment of APL cells. Thus, it would be interesting to carry out an in depth analysis on the role of ATG proteins on APL cell fate with the ultimate objective to identify new targetable pathways that overcome resistance to ATRA in APL patients. Likewise, the clinical significance of the pro-survival role of some ATG proteins in the development of differentiation syndrome deserves to be investigated 

## 5. Regulation of Autophagy in APL Cells Following Arsenic Trioxide Treatment

Similar to ATRA, the PML-RARα oncoprotein can be subjected to degradation by ATO through the autophagy machinery [14,77] (Figure 3). The contribution of autophagy in ATO-treated cells was first reported by Isakson et al., who showed that, in response to ATO, APL cells display upregulation of a subset of ATGs genes [14]. ATO promotes the co-localization of PML-RARα to the autophagy adaptor p62/SQSMT1, leading to its degradation through autophagy in APL cells [77]. Although there are multiple proteolytic pathways for the degradation of PML-RARα, autophagy seems to play a determinant role in ATO-induced clearance of PML-RARα [14]. From a molecular point of view, both ATO and ATRA inhibit MTOR, and subsequently decrease the phosphorylated form of the p70S6K protein [14,72]. In addition, depletion of ULK 1 prevents ATO- and ATRA-mediated degradation of PML-RARα in HeLa cells expressing PML-RARα, further supporting the conclusion that autophagy induced by both drugs is MTOR-dependent [14]. However, ATRA-induced autophagy occurs through a non-canonical autophagy pathway, as it does not require the BECN1 protein, whereas ATO-induced autophagy involves a mechanism that depends on BECN1 [14,72,80]. Moreover, a study by Ganesan et al. showed that proteasome inhibition along with ATO has an additive effect in inducing autophagy and promoting PML-RARα degradation [81]. Interestingly, this therapeutic combination is effective in relapsed APL patients. In parallel with these findings, another study demonstrated that autophagy is a key mechanism for anti-leukemic effects of ATO in AML cell lines and leukemic progenitors from AML patients, and this induction of autophagy flux is associated with mitogen-activated protein kinase kinase (MEK)/extracellular signal-regulated kinase (ERK)-dependent autophagic cell death [82]. The anti-leukemic effects of ATO and ATRA can be both mediated by targeting APL cells through ETosis, a type of cell death mediated by the release of neutrophil extracellular traps (ETs) [26,83]. APL cells and APL patients treated with moderate concentrations of ATO (0.5–0.75 μM) produce ETs and undergo ETosis via MTOR-dependent autophagy [26]. Moreover, a dynamic function of autophagy on the apoptotic effects of ATO has been also reported in human non-APL leukemia cell line HL-60 [84]. Indeed, pharmacological inhibition of autophagy prior to ATO treatment (1 h) enhances ATO-induced apoptosis, while autophagy inhibition after treatment with ATO (30 min) diminishes cell death. The authors concluded that autophagy has a pro-survival role in ATO-treated cells at the initiation stage, while it can kill the cells via apoptotic or autophagic cell death at the late stages [84]. On the other hand, it has been recently evidenced that microRNA-dependent control of autophagy contributes in microenvironment-mediated drug resistance to ATO [85]. In this content, miR-23a-5p represents an upstream factor in the upregulation of key autophagic genes, including AMBRA1, ATG2, ATG9 and ATG13. In fact, pharmacological inhibition of autophagy was shown to be able to overcome microenvironment-mediated ATO resistance implying the contribution of autophagy in resistance to treatment [85]. Under these conditions, whether or not microRNA-dependent control of autophagy contributes to the development of differentiation syndrome deserves to be investigated. 

## 6. Conclusions

While autophagy is downregulated in APL cells, this process can be activated in response to ATRA and ATO, two agents currently used in clinical settings. A body of evidence has revealed that autophagy is involved in differentiation and degradation of the PML-RARα oncoprotein in response to both ATRA and ATO treatment. Therefore, the induction of autophagy could be a desired strategy for the differentiation therapy of APL. In this line, rapamycin and lithium, two well-known activators of autophagy, may enhance therapeutic effectiveness of both ATRA and ATO in APL cells. We also showed that autophagy induced by nucleolar stress upon nucleostemin depletion can sensitize APL cell lines to differentiation therapy with ATRA [86]. Identifying the content of the autophagosomes that are degraded during differentiation should provide further insights into the molecular targets of autophagy during myeloid differentiation. 

However, there is another side of the coin, where autophagy, or more precisely some autophagy regulatory proteins, operate as pro-survival factors in response to differentiation agents, as well as to ATO treatment. This might ultimately be involved in the development of relapsed APL patients who are resistant to treatment and/or patients that manifest differentiation (ATRA) syndrome, due to the life-span extension of mature granulocytes. Moreover, autophagy has been shown to be involved in microenvironment-mediated drug resistance in AML cells, as its inhibition along with chemotherapy/ATO is potentially effective in the treatment of myeloid malignancies [85]. This suggests that induction of differentiation by autophagy is not solely sufficient to successfully clear the leukemic cells and provides the basis for using autophagy activators combined with cell death inducers in APL therapeutics. Finally, the data obtained from APL on the role and regulation of autophagy might be relevant to other malignancies, as both ATRA and ATO were shown to promote autophagy in non-APL cancer cells. 

## Figures and Tables

**Figure 1 ijms-20-03559-f001:**
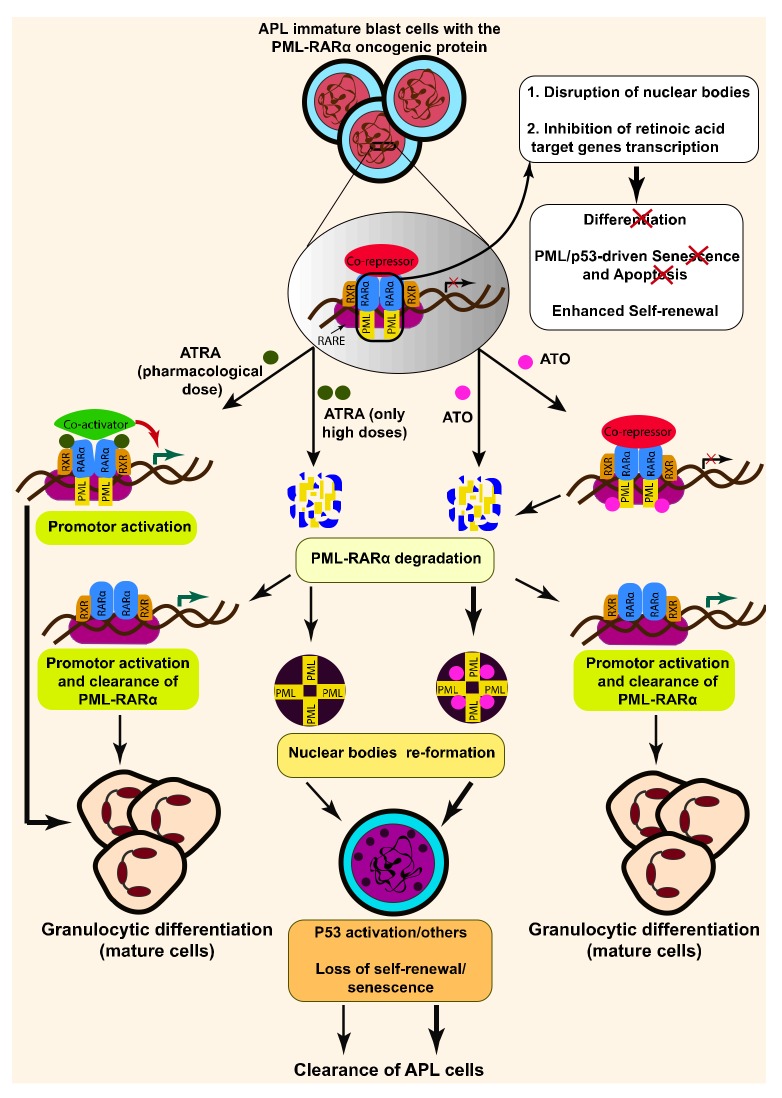
Mechanisms action of all-*trans* retinoic acid (ATRA) and arsenic trioxide (ATO) in promyelocytic leukemia retinoic acid receptor alpha gene (PML-RARα)-positive acute promyelocytic leukemia (APL) cells. In the absence of ATRA, heterodimers of retinoid X receptor (RXR)-RARα bind to retinoic acid response elements (RAREs) and recruit co-repressors and histone deacetylases, leading to the transcription repression of target genes. In APL cells, PML-RARα disrupts ATRA signaling by enhancing the binding of transcriptional repressor complexes. ATRA, at pharmacological doses, can bind to RARα moiety of PML-RARα and promote granulocytic differentiation of APL cells by restoring the transcription of retinoic acid target genes via the recruitment of transcription co-activator complexes to RAREs. High concentrations of ATRA can promote proteolysis of PML-RARα, leading to the re-formation of nuclear bodies and clearance of APL. ATO also induces partial differentiation and clearance of APL cells. This is mediated through binding of ATO to PML and PML-RARα, leading to sumoylation and poly-ubiquitination and subsequently proteoslysis of the PML-RARα oncoprotein.

**Figure 2 ijms-20-03559-f002:**
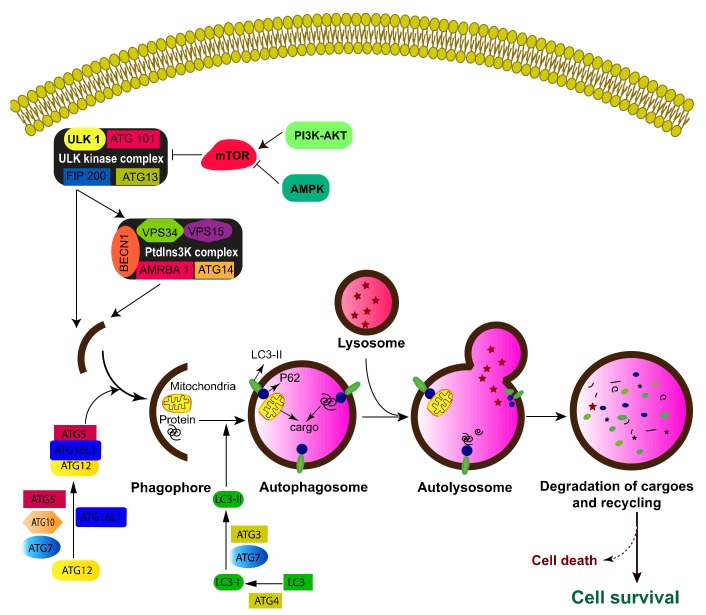
Molecular mechanisms of autophagy. Autophagy is orchestrated by a set of autophagy-related genes (ATGs) that are mostly under the control of the mechanistic target of rapamycin kinase (MTOR) and AMP-activated protein kinase (AMPK) signaling pathways. MTOR represses autophagy by suppressing the unc-51 like autophagy activating kinase 1/2 (ULK1/2) kinase complex activity while AMPK is an activator of the ULK kinase complex. The initiation step is started by the formation of a double membrane known as the phagophore, which required the assistance of two complexes, including the ULK1/focal adhesion kinase family interacting protein of 200 kD (Fip200) complex and the class III phosphoinositide 3-kinase (PI3K)/BECN 1 complex. The elongation step is mediated through the recruitment of two conjugation complexes, including the ATG5-ATG12-ATG16L1 complex and the ATG8 (LC3) complex. Upon autophagy, LC3 is first cleaved by ATG4 to produce LC3-I. The latter is recruited to the autophagosomal membrane upon its conjugation to phosphatidylethanolamine. The last step is the degradation of the contents of the autophagosome by the lysosomal acid hydrolases upon autophagosome–lysosome fusion. The autophagy adaptor proteins (e.g., p62/ sequestosome 1 (SQSMT1)) interact with specific cargoes directly or through their ubiquitin binding domain, allowing the deliverance of cargoes to autolysosomes for degradation.

**Figure 3 ijms-20-03559-f003:**
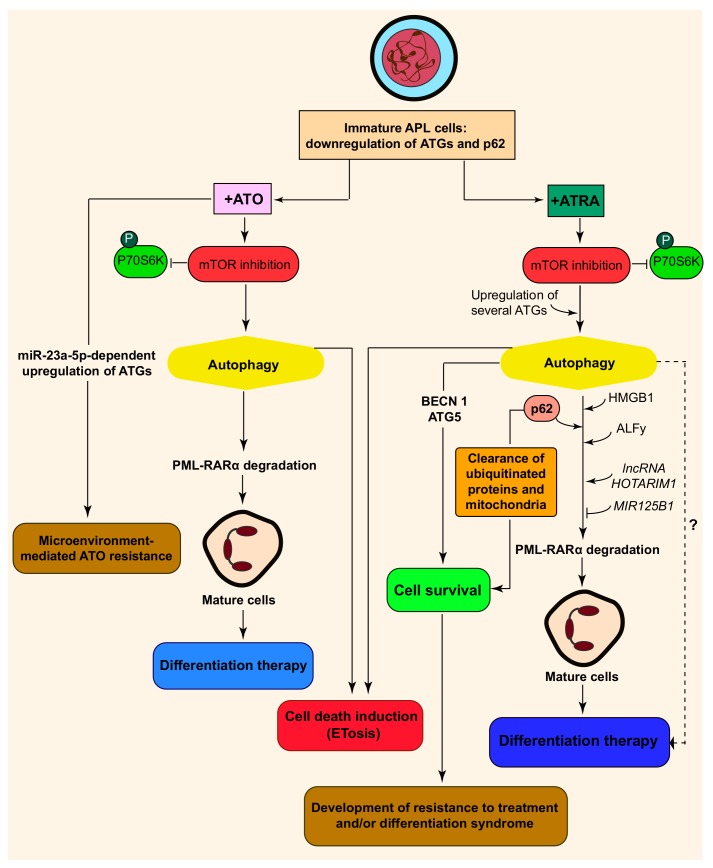
Regulation of autophagy in APL cells. The mRNA expression levels of several autophagy related genes are downregulated in APL immature blast cells relative to terminally differentiated granulocytes. ATO and ATRA are two anticancer agents used in clinical settings for the treatment of APL. Both agents induce autophagy through inhibition of MTOR and of ribosomal protein S6 kinase beta-1 (p70S6K), leading to the degradation of the PML-RARα oncoprotein, thus allowing the establishment of granulocyte differentiation of APL cells. In addition, both ATRA and ATO can activate ETosis, a type of cell death mediated by the release of neutrophil extracellular traps (ETs). Unlike ATRA, ATO-induced autophagy relies on BECN1. However, some autophagy regulatory proteins (e.g., BECN1 and p62/SQSTM1) act as pro-survival factors in APL cells subjected to ATRA treatment. This might contribute to the development of resistance to treatment and of differentiation syndrome in APL patients. Moreover, several ATG genes are upregulated in response to ATO through a microRNA-dependent mechanism. This constitutes another mechanism that might contribute to drug resistance in APL.

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
