# Peer review of "Autophagy: New Insights into Mechanisms of Action and Resistance of Treatment in Acute Promyelocytic leukemia"

_ijms, 2019, doi:10.3390/ijms20143559_

Reviewer 1 Report

This is a good review related to the impact of autophagy on the mechanisms of action of ATRA and ATO in acute promyelocytic leukemia. The content is appropriate, although I have some comments:

On the Introduction, I would include a figure of the oncogenic mechanisms of PML-RAR, in order to understand better the effects of the treatments. 

Another figure related to the regulation of autophagy in APL cells will be very useful to understand the other figure related to the treatments. You can combine the two figures. 

Thank you very much for your interesting manuscript. 

Author Response

Thanks for the suggestions

As requested by the referee, we includedc the oncogenic mechanism of PML-RARA in the figure 1 (please see white boxes in the attached figure 1)

As requested by the referee, we  added « the regulation of autophagy in APL » in the figure 3 ( please see attached figure 3 "downregulation of ATG proteins and p62".

thank you

Reviewer 2 Report

This is a well written and timely review on autophagy in APL.   The contrasts and similarities between the two therapeutic approaches to treatment using ATRA or Arsenic are well described and illustrated. 

My only comment would be that whilst the authors mention a discussion on the role ot autophagy in resistance - it is not a clear section in the same way that differentiation syndrome is covered. 

Author Response

Thanks for the suggestions

In fact, little is known  in the literature on the  role of autophagy and ATG proteins in  the development of differentiation syndrome in APL  patients.

This aspect was added in the manuscript text in the lines 320-322: “Likewise, the clinical significance of the pro-survival role of some ATG proteins in the development of differentiation syndrome deserves to be investigated” and in the lines  358-359:” Under this condition, whether or not microRNA-dependent control of autophagy contributes to the development of differentiation syndrome warrants to be investigated”.

thank you